# Cavefish cope with environmental hypoxia by developing more erythrocytes and overexpression of hypoxia-inducible genes

Corine M van der Weele, William R Jeffery*

Department of Biology, University of Maryland, College Park, United States

**Abstract** Dark caves lacking primary productivity can expose subterranean animals to hypoxia. We used the surface-dwelling (surface fish) and cave-dwelling (cavefish) morphs of *Astyanax mexicanus* as a model for understanding the mechanisms of hypoxia tolerance in the cave environment. Primitive hematopoiesis, which is restricted to the posterior lateral mesoderm in other teleosts, also occurs in the anterior lateral mesoderm in *Astyanax*, potentially pre-adapting surface fish for hypoxic cave colonization. Cavefish have enlarged both hematopoietic domains and develop more erythrocytes than surface fish, which are required for normal development in both morphs. Laboratory-induced hypoxia suppresses growth in surface fish but not in cavefish. Both morphs respond to hypoxia by overexpressing *hypoxia-inducible factor 1* (*hif1*) pathway genes, and some *hif1* genes are constitutively upregulated in normoxic cavefish to similar levels as in hypoxic surface fish. We conclude that cavefish cope with hypoxia by increasing erythrocyte development and constitutive *hif1* gene overexpression.

## Editor's evaluation

We believe it will have a significant impact on our understanding of cavefish adaptation in particular and adaptation to low oxygen environments in general.

*For correspondence:
Jeffery@umd.edu

Competing interest: The authors declare that no competing interests exist.

## Introduction

A unifying theme of cave habitats is complete darkness (*Culver and Pipan, 2009*) and the absence of primary productivity can subject cave-dwelling animals to oxygen deficiency (*Malard and Hervant, 2001*). We have used the teleost *Astyanax mexicanus*, which consists of surface-dwelling (surface fish) and cave-dwelling (cavefish) morphs, as a model to study adaptation to hypoxia in caves. Surface fish colonized subterranean waters in the Sierra de El Abra region of Mexico (*Mitchell et al., 1977*; *Gross, 2012*) during the late Pleistocene (*Fumey et al., 2018*; *Herman et al., 2018*), and their cavefish descendants evolved numerous traits for adaptation to perpetual darkness. Cavefish adaptations include more taste buds and cranial neuromasts, increased olfactory capacity, and fat deposits, which could facilitate feeding and survival during times of low food input, and decreased metabolic rates and loss of eyes, which may be important in energy conservation (*Yamamoto et al., 2004*; *Moran et al., 2014*; *Moran et al., 2015*; *Varatharasan et al., 2009*; *Blin et al., 2018*; *Yoshizawa et al., 2014*; *Hüppop, 1986*; *Xiong et al., 2018*). Hypoxia may be a deterrent to the survival of colonizing teleosts in the Sierra de El Abra, where oxygen reductions of 50% or more have been measured in cave waters (*Boggs and Gross, 2021*; *Rohner et al., 2013*; *Ornelas-García et al., 2018*). *Astyanax* cavefish thrive under these conditions, but little is known about how they have adapted to hypo-oxygenated waters.

Oxygen is bound to hemoglobin and transported through the body by red blood cells. Increased erythrocyte production is a well-documented strategy for coping with hypoxia at high altitude (*Haase, 2013*), but the role of red blood cells in adaptation to hypoxic caves is unknown. Hematopoiesis, the process in which erythrocytes and other blood cells develop during embryogenesis, occurs in two waves in vertebrates: the primitive and definitive waves (*Davidson and Zon, 2004*; *Paik and Zon, 2010*). In the zebrafish *Danio rerio* and other teleost embryos, the first or primitive wave occurs in the anterior and posterior lateral mesoderm beginning during the tailbud stage. The anterior lateral mesoderm (ALM) undergoes myelopoiesis to form macrophages, and some of these later differentiate into neutrophils and microglia (*Herbomel et al., 1999*; *Herbomel et al., 2001*; *Le Guyader et al., 2008*), whereas the posterior lateral mesoderm (PLM) undergoes erythropoiesis to form primitive erythrocytes. Subsequently, the PLM converges into the intermediate cell mass, where the precursors of endothelial cells lining the first blood vessels, primitive erythrocytes, and myeloid cells will ultimately differentiate. The second or definitive wave of hematopoiesis begins later in development when hematopoietic stem and progenitor cells are formed in the aorta-gonad-mesonephros (AGM), which seed the caudal hematopoietic tissue and form the posterior blood islands, where definitive erythrocytes begin to differentiate (*Paik and Zon, 2010*; *Gore et al., 2018*). Eventually, definitive hematopoiesis moves to the thymus and head kidneys in adults. The primitive and definitive waves of hematopoiesis are potential targets for adaptive changes induced by hypoxia.

Hematopoiesis is directed by a series of transcription factors producing different types of blood cells (*Davidson and Zon, 2004*; *Carroll and North, 2014*). The *growth factor independence 1* (*gfi1aa*) gene, which encodes a transcriptional repressor of many different genes (*Möröy et al., 2015*), is important for primitive hematopoiesis at several developmental stages (*Wei et al., 2008*; *Cooney et al., 2013*; *Moore et al., 2018*). In zebrafish, *gfi1aa* expression occurs throughout the period of embryonic segmentation, first in the PLM, then in the intermediate cell mass, and finally in the AGM (*Wei et al., 2008*; *Cooney et al., 2013*), where it directs the development of hemogenic endothelia into hematopoietic stem and progenitor cells, which ultimately differentiate into red blood cells (*Moore et al., 2018*). The *gfi1aa* gene is also expressed in the ALM, albeit weakly and only for a short time (*Wei et al., 2008*). Its expression at different stages of hematopoiesis makes *gfi1aa* an excellent marker for investigating differences between surface fish and cavefish hematopoiesis. The *LIM domain only 2* gene encodes another transcription factor essential for determination of hematopoietic domains early in primitive hematopoiesis (*Patterson et al., 2007*) and is also an excellent marker for differences during the initial stages of hematopoiesis.

Oxygen is required in aerobic organisms to produce energy, and insufficient oxygen leads to the activation of an evolutionarily conserved transcriptional response (*Majmundar et al., 2010*). The responses to oxygen deprivation are coordinated by activation of the hypoxia-inducible factors (HIF1, HIF2, and HIF3) (*Semenza and Wang, 1992*), a family of transcription factors controlling a large number of downstream target genes involved in promoting different responses to hypoxia (*Pelster and Egg, 2018*; *Rashid et al., 2017*). Increased expression of the α subunit of HIF1 and the HIF regulated genes are indicators of hypoxia (*Mandic et al., 2021*). Notable responses include the improvement of oxygen supply through increasing red blood cell production and function (*Haase, 2013*; *Kulkarni et al., 2010*; *Cai et al., 2020*), metabolic homeostasis balancing ATP production with oxygen consumption, management of Reactive Oxygen Species (ROS) via glycolysis, modulating anaerobic respiration and the pentose phosphate pathway (PPP) (*Singh et al., 2017*; *Wheaton and Chandel, 2011*; *Stincone et al., 2015*), and conservation of energy by growth inhibition (*Kamei, 2020*). *Astyanax* cavefish (*Hüppop, 1986*; *Moran et al., 2014*; *Aspiras et al., 2015*) and other cave-adapted animals (*Bishop et al., 2004*) have evolved reduced metabolic rates to conserve energy in the cave environment. However, the strategies used to conserve energy in hypoxic cave environments are poorly understood.

The present investigation shows that cavefish embryos develop significantly more erythrocytes than surface fish. The results also revealed that cavefish larvae have constitutively high expression of some of the HIF1 pathway genes and show no effects of oxygen depletion on growth, traits that could promote larval survival in hypoxic environments. These studies reveal that changes in development of the circulatory system may underlie adaptation of cavefish to hypo-oxygenated environments.

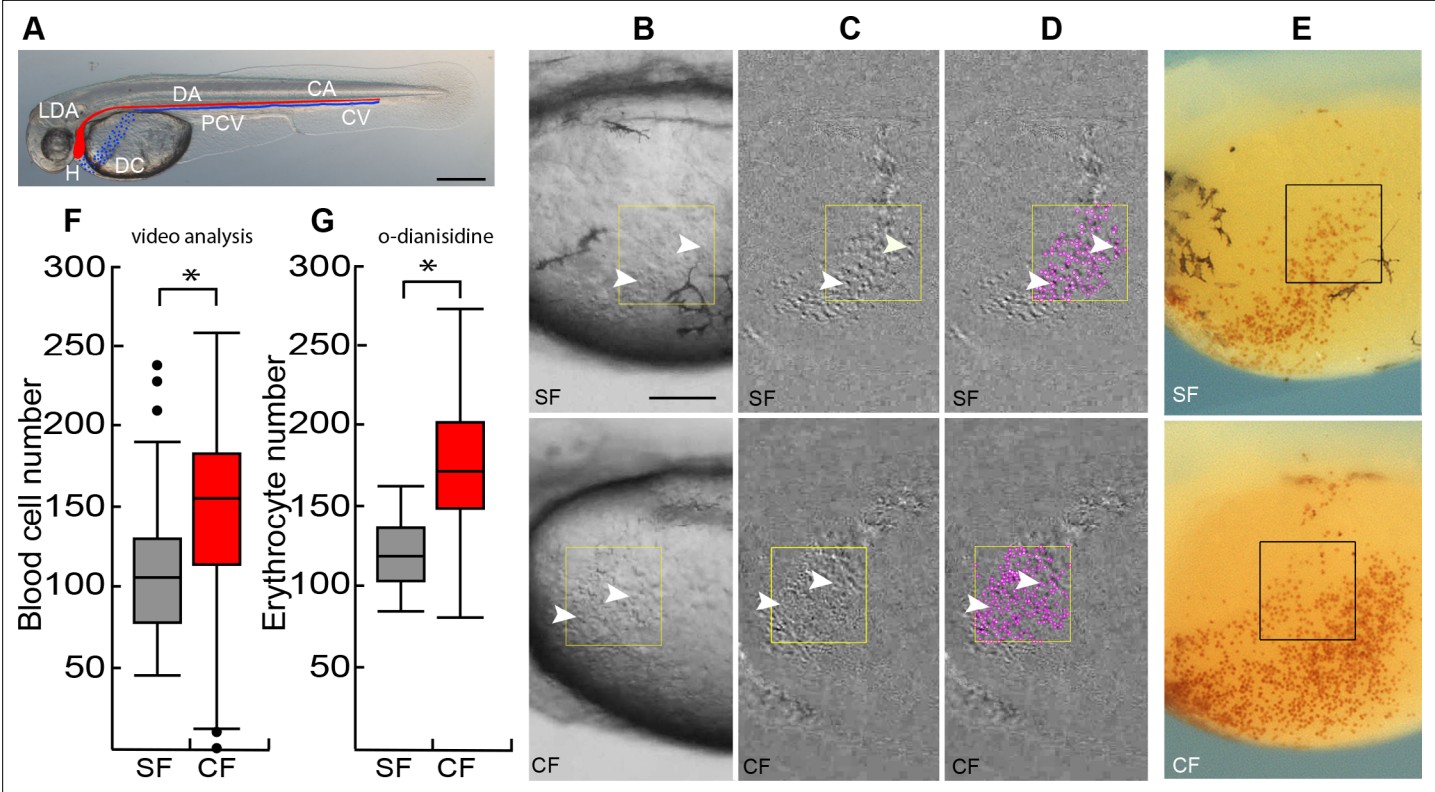

**Figure 1.** Cavefish have more erythrocytes than surface fish. (**A**) At 36 hpf blood circulates from the heart (H) through the lateral dorsal aorta (LDA), the dorsal aorta (DA) and caudal artery (CA) (blood flow shown in red), and back to the heart through the caudal vein (CV), the posterior cardinal vein (PCV), and the Duct of Cuvier (DC) (blood flow shown in blue). Scale bar is 200 μm. (**B–D**). Video analysis of circulating blood cells (arrowheads) in a region of interest (ROI, boxes) in the DC of surface fish (SF) and cavefish (CF) larvae at 36 hpf visualized by subtraction of two video frames 0.15 s apart (**C, D**) and quantified using the plugin TrackMate (ImageJ) (D, purple dots). (**E**) o-dianisidine staining of red blood cells in the DC of surface fish (SF) and cavefish (CF) larvae at 36 hpf. Boxes: erythrocyte quantification regions. Scale bar in E is 100 μm, B-E are the same magnifications. (**F, G**). Quantification of blood cells in the DC by video analysis (**F**) and o-dianisidine staining. Box plots show the median, quartiles, min-max values and outliers (dots). Asterisks: $p < 0.05$, N = 72 (in F) and $p < 0.0001$, N = 36 (in G). Statistics by Wilcoxon/Kruskal-Wallis Rank Sums test.

The online version of this article includes the following source data for figure 1:

**Source data 1.** Cavefish have more erythrocytes than surface fish.

## Results

### Cavefish have more erythrocytes than surface fish

To test the hypothesis that cavefish develop more red blood cells as a response to environmental hypoxia, we compared erythrocytes between surface fish and cavefish embryos in several different ways. First, we conducted direct observations of blood cell numbers after circulation begins in the Duct of Cuvier (DC) at 36 hpf (*Figure 1A–D*). At this time, the circulatory system is still partially open, and blood cells can be imaged or stained and quantified as they flow across the yolk (*Videos 1 and 2* and *Figure 1B–D and F*). The number of circulating blood cells was quantified in sequential video frames using Image-J analysis software, and the results showed that cavefish have more blood cells than surface fish (*Figure 1B–D and F*). Second, the number of erythrocytes was quantified in the DC by staining with o-dianisidine (*Iuchi and Yamamoto, 1983*),

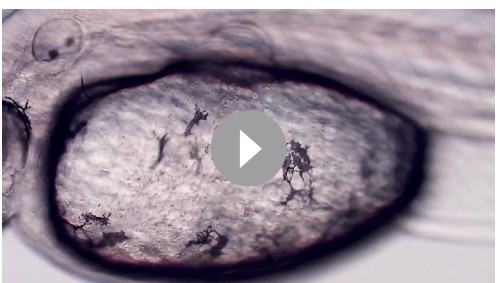

**Video 1.** Blood cells flowing from the cardinal vein through the Duct of Cuvier to the heart in surface fish.
https://elifesciences.org/articles/69109/figures#video1

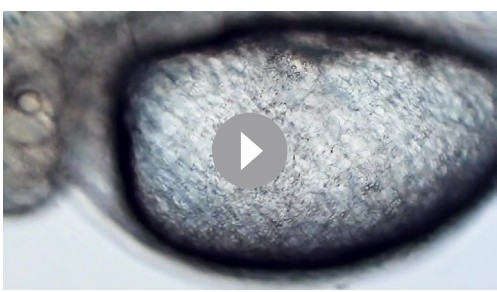

**Video 2.** Blood cells flowing from the cardinal vein through the Duct of Cuvier to the heart in cavefish. https://elifesciences.org/articles/69109/figures#video2

and the results revealed that cavefish have more erythrocytes than surface fish (*Figure 1E and G*). Third, expression of the *hbb2* and *hbbe2 ß-hemoglobin* genes was followed during surface fish and cavefish development by qPCR and in situ hybridization (*Figure 2A, B and D–F*). The qPCR results revealed significantly higher levels of *hbb2* mRNA at 24 hpf and 60 hpf and *hbbe2* mRNA at 60 hpf (*Figure 2A and B*). In situ hybridization also showed more extensive *hbb2* staining in cavefish compared to surface fish (*Figure 2D–F*), which was particularly evident in the yolk mass at 36 hpf (*Figure 2E*). Fourth, the expression of *gfi1aa*, which encodes a transcription factor essential for hematopoiesis (*Cooney et al., 2013*), was compared by qPCR and in situ hybridization (*Figure 2C and G*). The results indicated that *gfi1aa* mRNA levels were significantly increased in cavefish compared to surface fish at 24 hpf (*Figure 2C and G*), prior to detection of the highest levels of hemoglobin transcription (*Figure 2A and B*). These results provide multiple lines of evidence indicating that red blood cells are increased during cavefish development.

## Hematopoietic domains in surface fish and cavefish embryos

To determine the developmental basis of erythrocyte enhancement, we compared the timing and spatial expression of *hbb2*, *gfi1aa*, and other hematopoietic marker genes during primitive hematopoiesis in surface fish and cavefish embryos (*Figure 3*). As in zebrafish (*Brownlie et al., 2003*; *Wei et al., 2008*), *hbb2* was expressed in the intermediate cell mass of the PLM during primitive hematopoiesis in *Astyanax* surface fish and cavefish embryos (*Figure 3A*). In striking contrast to zebrafish, in which erythropoiesis occurs exclusively in the PLM, *hbb2* expression was detected in both the ALM and PLM of *Astyanax* surface fish and cavefish embryos (*Figure 3A*). The expression of *gfi1aa* was also present in the ALM and PLM during *Astyanax* embryogenesis (*Figure 3A*). Furthermore, *gfi1aa* and *hbb2* staining were stronger in the cavefish ALM and PLM compared to surface fish (*Figure 3A*). To further evaluate the primitive erythropoietic domains, we examined expression of the *lim domain only 2* (*lmo2*) gene, a marker of developing blood cells (*Patterson et al., 2007*), by in situ hybridization. The results confirmed blood cell development in both the ALM and PLM in *Astyanax* embryos and revealed that both primitive erythropoietic domains are expanded in cavefish embryos (*Figure 3C*).

In zebrafish, the ALM undergoes myelopoiesis, rather than erythropoiesis, producing macrophages that disperse throughout the embryo (*Herbomel et al., 1999*). Therefore, we asked if macrophage development in *Astyanax* is compromised by the expansion of primitive erythropoiesis into the ALM. To address this possibility, we examined expression of the *l-plastin1* (*lcp1*) gene, a marker for macrophage differentiation (*Herbomel et al., 2001*). In situ hybridization showed that *lcp1* expressing cells were distributed throughout the ventral body and yolk masses of surface fish and cavefish embryos, implying that macrophages migrated from the ALM and differentiated (*Figure 3B*). Therefore, macrophage development does not appear to be compromised by the expansion of erythropoiesis into the ALM.

In summary, the results reveal the presence of a novel domain of primitive erythropoiesis in the anterior region of *Astyanax* embryos and suggest that red blood cells are increased in cavefish due to expansion of both the anterior and posterior primitive erythropoiesis domains.

## Blood cell enhancement is a maternally controlled trait

Hematopoietic domains develop in the lateral mesoderm of zebrafish embryos during gastrulation (*Kimmel et al., 1990*), and changes in gastrulation and axis determination are under maternal control in cavefish (*Ma et al., 2018*; *Torres-Paz et al., 2019*). Therefore, to determine whether the increase in cavefish blood cells is controlled by maternal or zygotic changes, we conducted reciprocal hybrid crosses (cavefish female x surface male and surface fish female x cavefish male) and quantified blood cells in the DC of F1 hybrid embryos by video imaging (*Figure 4*). As controls, crosses were also done

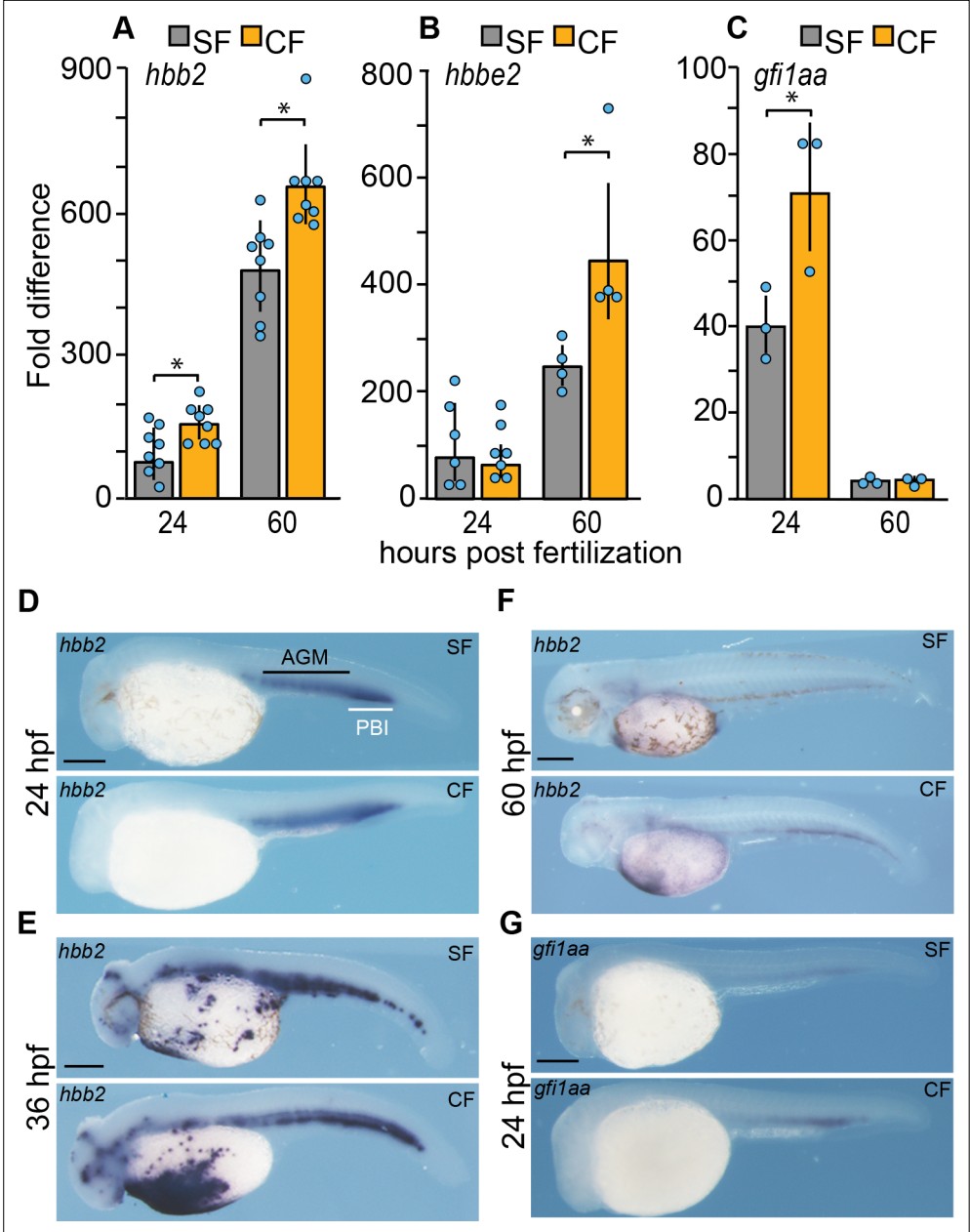

**Figure 2.** Expression of hematopoietic genes during surface fish and cavefish development. (**A–C**). qPCR quantification of *hbb2*, *hbbe2*, and *gfi1aa* transcripts in surface fish (SF) and cavefish (CF) larvae at 24 hpf and 60 hpf. Error bars: range of fold change. Asterisks: (**A**) N = 8. p = 0.0045 for SF vs CF at 24 hpf and p = 0.0254 for SF vs CF at 60 hpf; (**B**) N = 4. p = 0.0271 for SF vs CF at 60 hpf.; (**C**) N = 3. p = 0.0413 for SF vs CF at 24 hpf. Statistics by two-way ANOVA followed by Student's t test. (**D–G**). In situ hybridizations showing *hbb2* staining at 24, 36, and 60 hpf, (**D–F**) and *gfi1aa* staining (**G**) at 24 hpf in CF and SF. AGM: aorta-gonad-mesonephros. PBI: posterior blood island. Scale bars are 200 µm in each frame. Magnifications are the same in SF and CF.

The online version of this article includes the following source data for figure 2:

**Source data 1.** Expression of hematopoeitic genes during surface fish and cavefish development.

and blood cell numbers quantified in the offspring of parental cavefish x cavefish and surface fish x surface fish crosses. The F1 hybrid progeny of the cavefish female x surface fish male and the cavefish x cavefish crosses (*Figure 4C and D*) developed more blood cells than the progeny of surface fish female x cavefish male and surface fish x surface fish crosses (*Figure 4A and B*), showing that the

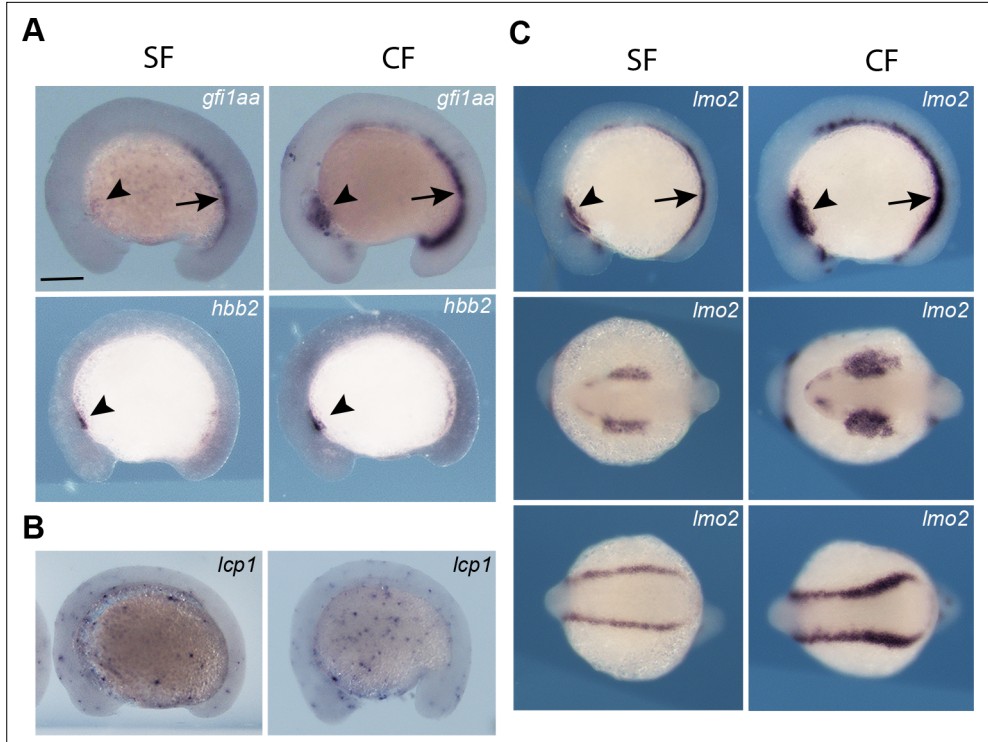

**Figure 3.** Hematopoietic domains in 14 hpf surface fish and cavefish embryos. (**A**). In situ hybridizations showing expression of the *gfi1aa* (top frames) and *hbb2* (bottom frames) genes in the anterior lateral mesoderm (ALM, arrowheads) and posterior lateral mesoderm (PLM, arrows) in surface fish (SF) and cavefish (CF). (**B**) Expression of the macrophage marker gene *lcp1* in SF and CF embryos. (**C**). Expression of the hematopoietic marker gene *lmo2* in the ALM and PLM of CF and SF embryos, and expansion of expression in CF embryos. Scale bar in A is 200 μm; magnification is the same in all frames.

extent of erythrocyte development is dependent on the source of eggs. These results reveal that the increase in cavefish blood cells is a maternally controlled trait.

## Erythrocytes are required for Astyanax development

Erythrocytes also develop early in zebrafish embryogenesis, but are not required for oxygen delivery until much later in development (*Pelster and Burggren, 1996*). Accordingly, we performed experiments to determine whether red blood cells have a role in early *Astyanax* development. To address this question, we compared the effects of hemolytic anemia in surface fish and cavefish embryos (*Figure 5*). Hemolysis of red blood cells was induced by treatment with phenylhydrazine (PHZ) (*Houston et al., 1988*) beginning at 14 hpf (13–17 somite stage), and *hbb2* staining, red blood cell number, and post-anal tail length (used as a proxy for normal growth and development) were compared in surface fish and cavefish after 14 hr of PHZ exposure (*Figure 5A–C*). We found that *hbb2* staining and red blood cells were reduced at increasing PHZ concentrations in both surface fish and cavefish embryos (*Figure 5A and B*). However, at the same PHZ concentration both erythrocyte phenotypes were more strongly affected in surface fish than in cavefish, consistent with more red blood cells in the latter. The results also showed that growth decreased in concert with hemoglobin transcript and erythrocyte reduction in surface fish, and concordant with more red blood cells, no effects on growth were seen in PHZ-treated cavefish (*Figure 5B*). The PHZ-treated embryos developed axial defects, most notably edemas, twisted notochords, and shortened and ventrally bent tails (*Figure 5C*), which were more prevalent at lower PHZ concentrations in surface fish than cavefish embryos (*Table 1*). To address the possibility that reduced tail length was a consequence of abnormal notochord morphogenesis rather than growth suppression, tail length and notochord morphology were investigated at 36 hpf in surface fish exposed to a PHZ concentration that virtually abolished red blood cells (*Figure 5D*). Under these conditions, post-anal tail growth was significantly reduced, but notochord defects were not detected,

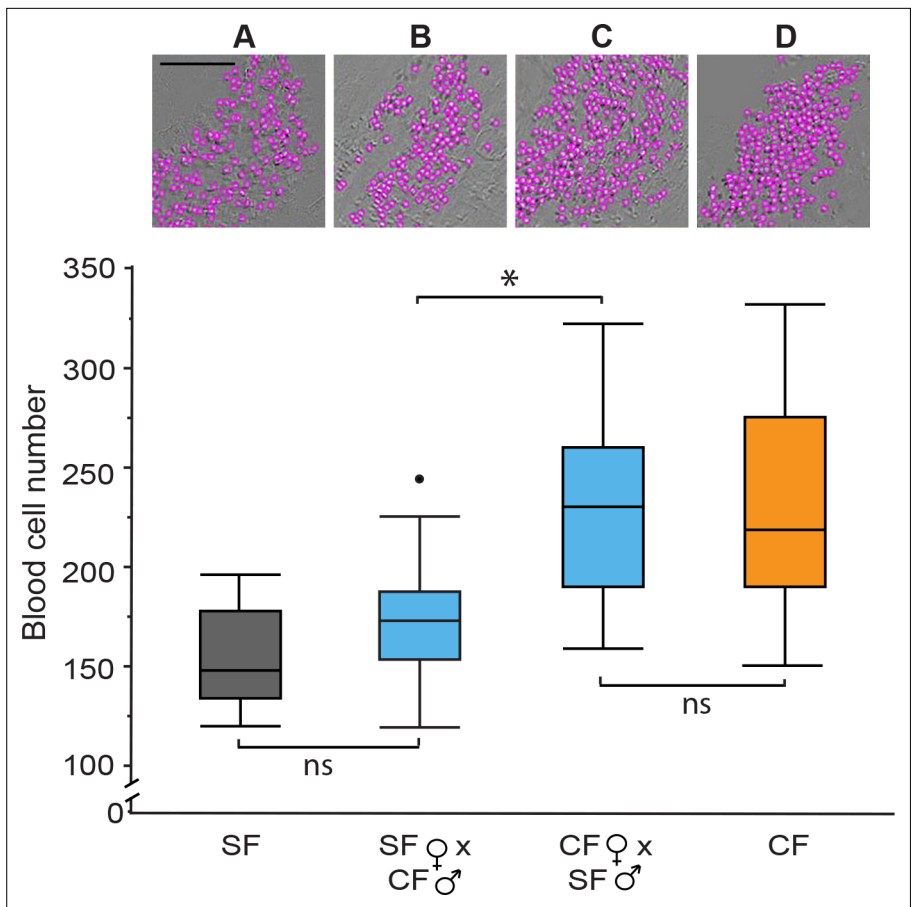

**Figure 4.** Maternal control of increased blood cells in cavefish determined by reciprocal hybridizations. Video imaging in the Duct of Cuvier and blood cell quantification at 34 hpf in the F1 progeny of a (**A**) surface fish (SF) X SF control cross, (**B**) a SF female X cavefish (CF) male cross, (**C**) a CF female X SF male cross, and (**D**) a CF X CF control cross. (**A–D**) Top row. Representative images of blood cell number aligned with box plots below. Scale Bar is 100 µm; magnifications are the same in each frame. Bottom row. Boxplots of blood cell numbers showing medians, quartiles, min-max values, and outliers (dots). Asterisks: p < 0.05. ns: not significant. N = 20 for each box plot. Statistics by Wilcoxon/Kruskal-Wallis Rank Sums test followed by Wilcoxon for each pair.

The online version of this article includes the following source data for figure 4:

**Source data 1.** Maternal control of increased blood cells in cavefish determined by reciprocal hybridizations.

suggesting that hemolytic effects on axial growth are related to the absence of red blood cells. The results show that erythrocytes are required during early development in *Astyanax* and more red blood cells may reduce the sensitivity of cavefish to hemolytic anemia.

## Cavefish but not surface fish maintain growth in a hypoxic laboratory environment

Subterranean waters harboring *Astyanax* cavefish are low in oxygen (*Rohner et al., 2013*; *Ornelas-García et al., 2018*; *Boggs and Gross, 2021*), which is essential for normal development and growth (*Giaccia et al., 2004*). Therefore, we hypothesized that cavefish may be more tolerant to the effects of hypoxia on growth than surface fish. To test this hypothesis, 36 hpf surface fish and cavefish were exposed to a hypoxic laboratory environment for 18 hr (*Figure 6A and B*). The effects on growth were determined by measuring post-anal tail length before and after hypoxia treatment, and the number of erythrocytes was quantified by o-dianisidine staining at the end of the treatment period. The results showed that hypoxia reduced the number of red blood cells in the DC and suppressed growth in surface fish but not cavefish (*Figure 6B*), showing that cavefish are more resistant to oxygen depletion. Reasoning that eye growth would be especially sensitive to hypoxia because of the energy

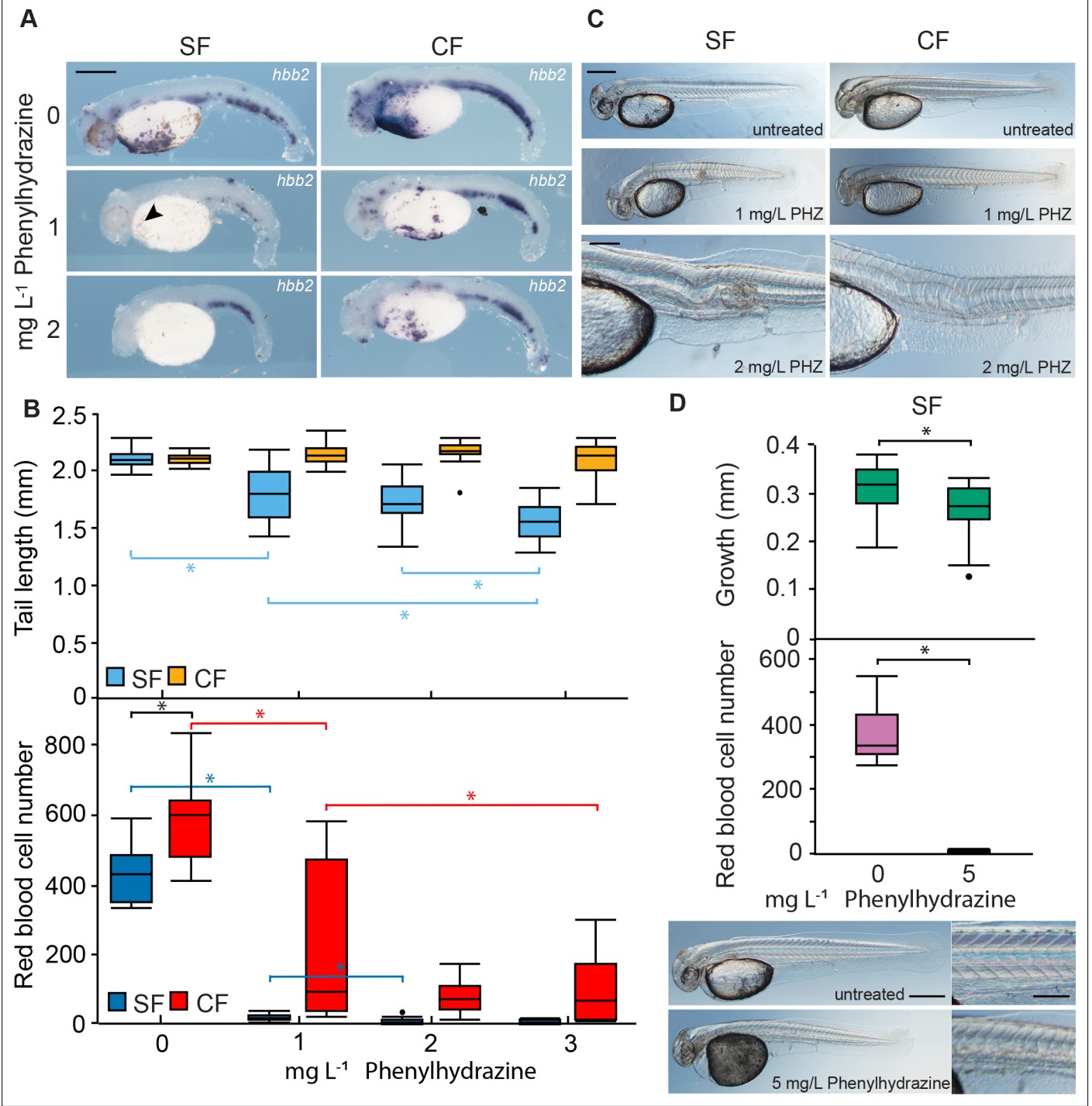

**Figure 5.** Differential effects of hemolytic anemia on surface fish and cavefish development. (**A**). In situ hybridization with the *hbb2* gene marker in 40 hpf larvae treated with different phenylhydrazine (PHZ) concentrations. Magnification is the same in all frames. (**B**). Larval tail length (top) and erythrocyte number in the Duct of Cuvier (DC) (bottom) in SF and CF embryos as a function of PHZ concentration. Asterisks in B (top) p < 0.0001. N = 23 for SF 0, 2 mg L$^{-1}$ and CF 3 mg L$^{-1}$. N = 24 for SF 1 and 3 mg L$^{-1}$ and CF 0, 1, and 2 mg L$^{-1}$. Asterisks in B (bottom) consecutively from left to right: p = 0.0197, p < 0.0001, p = 0.0002, p < 0.0001, and p 0.0102. N = 20 for SF 0, 1, 2 and CF 3 mg L$^{-1}$. N = 10 for SF 3 mg L$^{-1}$. N = 19 for CF 0 mg L$^{-1}$. N = 21 for CF 1 mg L$^{-1}$. N = 22 CF 2 mg L$^{-1}$. Statistics by Wilcoxon/Kruskal-Wallis Rank Sums test followed by Wilcoxon for each pair. (**C**). Axial defects induced by hemolytic anemia in SF and CF embryo. Scale bars are 200 μm; magnifications are the same in top four frames and bottom two frames. Data from B and C were obtained from the same embryos. (**D**). Effects of a 1 hr. treatment with 5 mg/L PHZ on SF post-anal tail growth (top), erythrocyte number measured in the DC (middle), and axial development (bottom left and right). Scale bars in A, C (upper frame), and D (left frame) are 500 μm. Scale bars in C (lower frame) and D (right frame) are 200 μm. Asterisk (top): p = 0.0115, N = 20. Statistics by Student's t test. Asterisk (middle): p = 0.0001, N = 10. Statistics by

*Figure 5 continued on next page*

Figure 5 continued

Wilcoxon/Kruskal-Wallis Rank Sums test followed by Wilcoxon for each pair. Box plots in C and D show whiskers ( = 5%), mean (line) and outlier (dot). SF: surface fish. CF: cavefish. Box plots show the median, quartiles, min-max values and outliers (dots).

The online version of this article includes the following source data for figure 5:

**Source data 1.** Differential effects of hemolytic anemia on surface fish and cavefish development.

cost associated with retinal differentiation and maintenance (*Wong-Riley, 2010*; *Moran et al., 2015*), eye size was also examined in surface fish and cavefish exposed to normoxia or hypoxia for 18 hr. The results showed that eye size was more sensitive to hypoxia relative to normoxia in surface fish compared to cavefish (*Figure 6C*). The results suggest that cavefish growth is resistant to hypoxia and development of more erythrocytes may contribute to hypoxia tolerance.

## Differential expression of oxygen-sensitive genes during hypoxia

To determine the molecular responses of surface fish and cavefish to hypoxia, we quantified some of the key oxygen sensitive genes by qPCR (*Figure 7*). Exposure of surface fish or cavefish larvae to hypoxia for 2 hr. resulted in increased transcript levels of the *insulin growth factor binding protein 1* a (*igfbp1a*) gene, the HIF1 family genes *hif1aa* and *hif1a-like2*, and the HIF1-regulated downstream gene *hexokinase 1* (*hk1*) (*Figure 7*). In contrast, the *hifab* and *hif1a-like* genes were expressed at similar levels in normoxic cavefish and hypoxic surface fish (*Figure 7*), implying constitutive expression in cavefish. The expression of the *glucose-6-phosphate dehydrogenase* (*g6pd*), *lactate dehydrogenase a* (*ldha*), and *pyruvate dehydrogenase kinase 1* (*pdk1*) genes were not significantly increased by hypoxia in surface fish or cavefish (*Figure 7*). The results indicate that *igfbp1*, *hif1a-like2,* and *hk1* are upregulated by hypoxia in both surface fish and cavefish, whereas *hif1aa* and *hif1a-like* are expressed at higher levels in cavefish than in surface fish in either normoxic or hypoxic laboratory conditions. Thus, cavefish exhibit a complex molecular response to hypoxia: some oxygen sensing genes are upregulated, others are not changed, and some of the latter genes may be insensitive to oxygen depletion because they are already expressed at maximal levels in normoxic cavefish.

## Discussion

**Table 1.** Penetrance of abnormalities in phenylhydrazine-treated surface fish and cavefish embryos.

| Morph | PHZ concentration | % edema | % defective notochord |
|-------|-------------------|---------|-----------------------|
| SF | 0 | 0 | 0 |
| CF | 0 | 0 | 0 |
| SF | 1 | 4.1 | 87.5 |
| CF | 1 | 0 | 41.7 |
| SF | 2 | 75.0 | 100 |
| CF | 2 | 8.3 | 87.5 |
| SF | 3 | 95.8 | 100 |
| CF | 3 | 25.0 | 100 |

PHZ: phenylhydrazine. SF: surface fish. CF: cavefish. N = 24 for each morph treatment.

The online version of this article includes the following source data for table 1:

**Source data 1.** Penetrance of abnormalities in phenylhydrazine-treated surface fish and cavefish embryos.

We show that cavefish embryos develop more red blood cells than surface fish embryos, a classic response to hypoxia (*Martinez et al., 2004*; *Timmerman and Chapman, 2004*; *Rutjes et al., 2007*; *Haase, 2013*), when they are raised under normoxic laboratory conditions. This conclusion is based on direct counts of blood cells, including erythrocytes, increased expression of multiple erythropoietic marker genes, and hyperexpression of the *gfi1aa* gene, a crucial regulator of hematopoiesis. In addition, we report that the increase in erythrocytes may be caused by expansion of embryonic hematopoietic domains in cavefish compared to surface fish. Finally, we provide evidence that cavefish have evolved a permanent response to hypoxia involving the constitutive overexpression of the some of the *hif1* genes, which persists under normoxic laboratory conditions.

Hematopoiesis occurs in the primitive and definitive waves in vertebrates (*Davidson and Zon, 2004*; *Paik and Zon, 2010*). Increased *hbb2*, *hbbe2*, and *gfi1aa* expression was observed during embryogenesis and after larval hatching in cavefish. In zebrafish, transfusion experiments

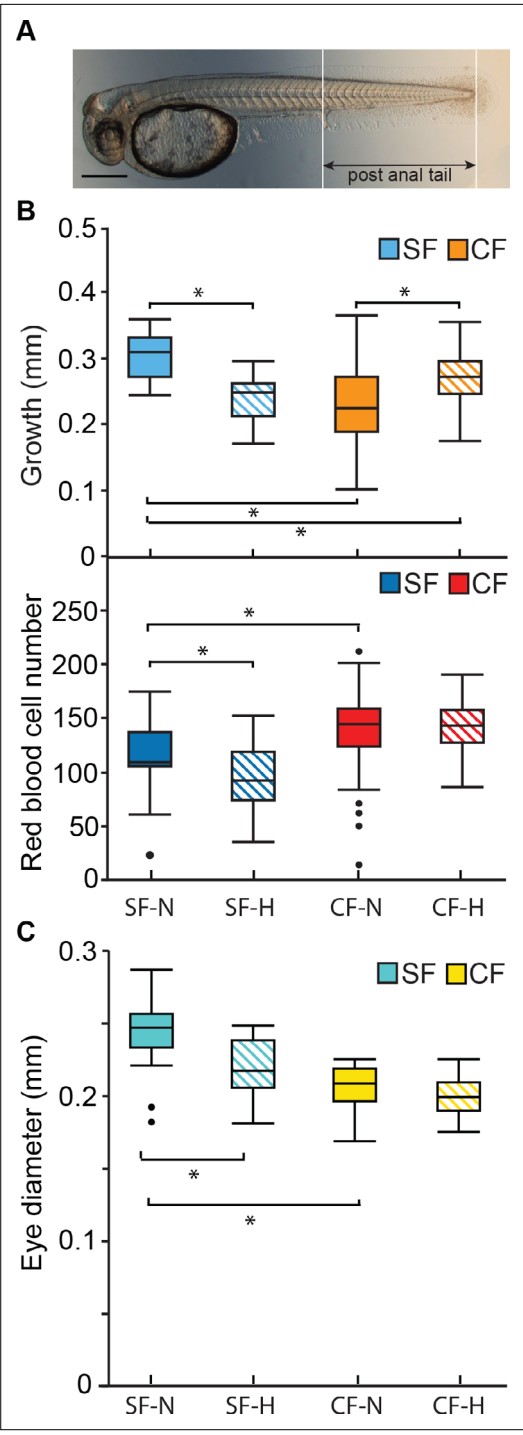

*Figure 6 continued*

Kruskal-Wallis Rank Sums test followed by Wilcoxon for each pair. Middle frame asterisks from left to right: p = 0.0116, and p = 0.0078. N = 22 for SF-N. N = 21 for SF-H. N = 47 for CF-N, and N = 45 for CF-H. Statistics by one-way ANOVA followed by Student's t-test. (**C**) Difference in eye diameters in surface fish and cavefish after 18 hr. of hypoxia or normoxia. Bottom frame asterisks: p = 0.0002 and p < 0.0001. N = 18 for SF-N. N = 14 for SF-H. N = 30 for CF-N. N = 28 for CF-H. Statistics by one-way ANOVA followed by Student's t-test. Box plots show median, quartiles and min-max values.

The online version of this article includes the following source data for figure 6:

**Source data 1.** Effects of laboratory induced hypoxia on surface fish and cavefish growth.

indicate that blood cells circulating during the first four days of development are primitive erythrocytes (*Weinstein et al., 1996*). Accordingly, our findings suggest that *Astyanax* larval blood cells are primitive erythrocytes derived from the first wave of hematopoiesis. Nevertheless, we also found stronger expression of erythropoietic marker genes in the AGM and caudal hematopoietic tissue, which are involved in definitive hematopoiesis, implying that the second wave may also be expanded in cavefish. Further investigation will be required to determine whether erythrocytes are also increased in adults, although this possibility seems likely because adult cavefish have increased levels of hemoglobin transcripts (*Sears et al., 2020*) and more erythrocytes in their head kidneys compared to surface fish (*Peuß et al., 2020*). We conclude that cavefish have evolved a permanent enhancement in red blood cells as a potential adaptation for survival in hypoxic cave waters.

The results showed that surface fish and cavefish primitive erythrocytes are derived from two different hematopoietic domains, one located anteriorly in the ALM and the other posteriorly in the PLM. Importantly, both the anterior and posterior embryonic domains show expanded marker gene expression in cavefish compared to surface fish. It is noteworthy that expansion of the posterior hematopoietic domain can also be induced in zebrafish by overexpressing LMO2 together with its partner transcription factor SCL/TAL1 (*Gering et al., 2003*), suggesting that teleosts have the capacity to modulate blood cell numbers by changing the size of primitive erythropoietic domains during development. The existence of erythrocyte forming capacity in the ALM

**Figure 6.** Effects of laboratory induced hypoxia on surface fish and cavefish growth. (**A**). Image of a 36 hpf surface fish larval showing post-anal tail length. Scale Bar is 200 µM. (**B**). Relative post-anal tail growth (top) and erythrocyte numbers (bottom) of surface fish and cavefish over an 18 hr. period under normoxic and hypoxic (1 mg/L oxygen) conditions. Top frame asterisks from left to right: p < 0.0001, p = 0.002; p < 0.0001, and p = 0.0021. N = 23 for SF-N and SF-H. N = 46 for CF-N. N = 47 for CF-H. Statistics by Wilcoxon/

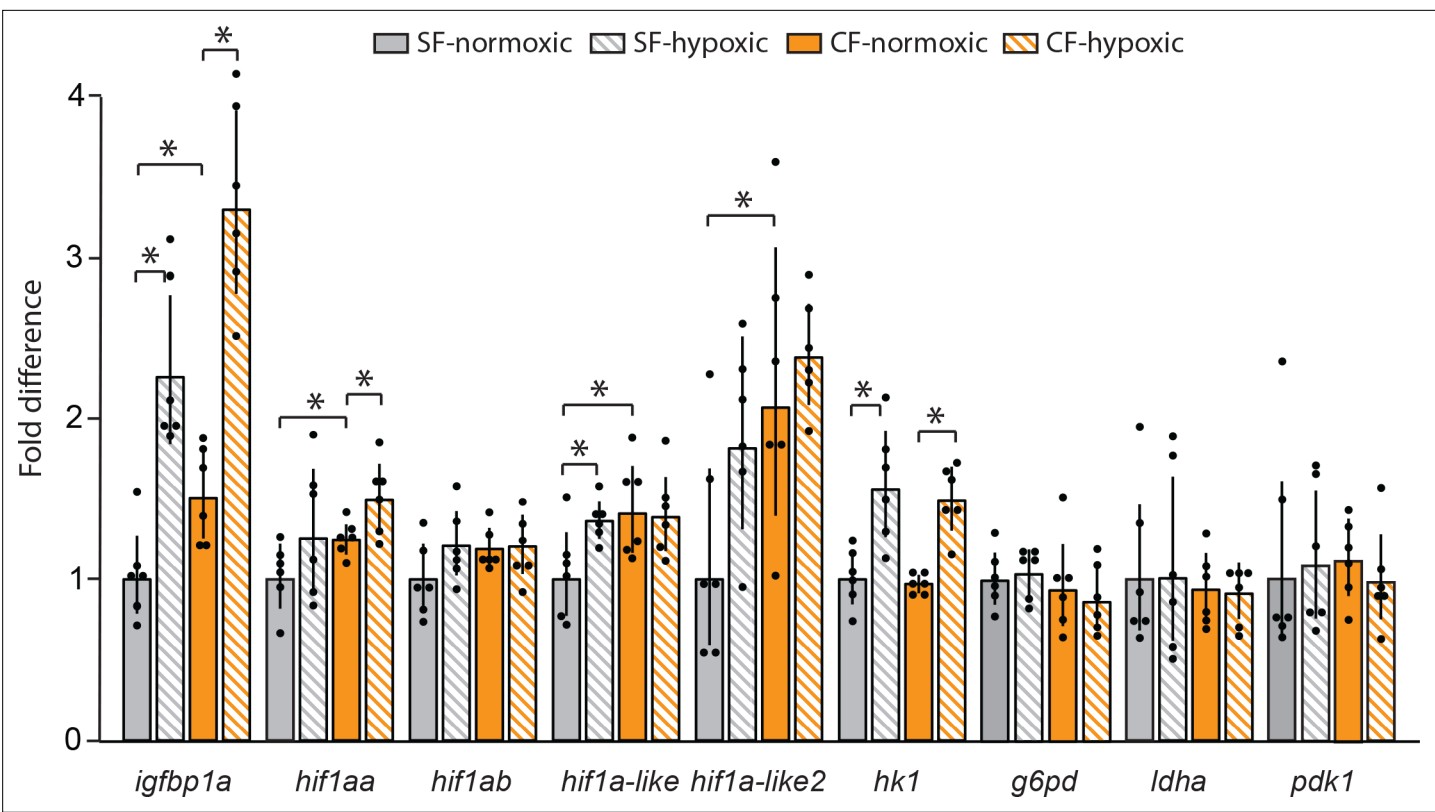

**Figure 7.** Changes in expression of some of the key HIF pathway genes in surface fish and cavefish exposed to hypoxia (1 mg/L oxygen) or normoxia. Bars indicate qPCR fold difference over surface fish in normoxic condition. Error bars: range of fold change. Asterisks from left to right: $p < 0.0001$, $p < 0.0001$, $p = 0.0044$, $p = 0.0306$, $p = 0.453$, $p = 0.0155$, $p = 0.0083$, $p = 0.0306$, $p = 0.0002$, and $p = 0.0003$. N = 6 for each determination. Statistics by one-way ANOVA followed by Student's t-test (*igfbp1a*, *hif1ab*, *hif1a-like*, *hif1a-like2*, *hk1*, *g6pd*, *ldha*, *pdk1*) or with unequal variance by Wilcoxon/Kruskal-Wallis Rank Sums test followed by Wilcoxon for each pair (*hif1aa*).

The online version of this article includes the following source data for figure 7:

**Source data 1.** Changes in expression of some of the key HIF pathway genes in surface fish and cavefish exposed to hypoxia (1 mg/L oxygen) or normoxia.

of surface fish, albeit smaller than in cavefish, suggests that the expansion of primitive erythropoiesis may pre-adapt *Astyanax* for colonizing hypoxic cave environments.

The expansion of erythropoiesis into the ALM during primitive hematopoiesis in *Astyanax* contrasts strikingly to zebrafish, medaka, and other teleosts that have been investigated, in which primitive erythropoiesis is confined to the PLM, and the ALM is devoted only to myelopoiesis (*Detrich et al., 1995*; *Govoni et al., 2005*; *Moriyama et al., 2010*). Expression of the macrophage marker *lcp1* (*Herbomel et al., 2001*) indicated that myelopoietic precursors had already differentiated into macrophages as erythrocyte progenitors were being formed in the ALM during early *Astyanax* development. Cell-tracing studies have shown that blood precursor cells can produce either erythrocytes or macrophages, but not both cell types (*Warga et al., 2009*), indicating an opposing relationship between the erythroid and myeloid cell populations. However, this relationship can be reversed by knockdown of the myeloid-controlling transcription factor PU.1, which induces blood precursors of the ALM to form erythrocytes instead of macrophages (*Rhodes et al., 2005*). Apparently, teleosts are potentially able to undergo erythropoiesis in the ALM and PLM, but in most species this capacity is normally confined to the PLM. In *Astyanax*, the balanced relationship between myeloid and erythroid cell specification appears to have tilted toward the production of erythrocytes. However, the development of erythrocytes in the ALM may not be at the expense of fewer myeloid cells because there is also an overall expansion of the ALM (and PLM), which may promote the specification of more blood precursor cells. In future studies, it will be interesting to determine how the expansion of erythropoietic potential is controlled in cavefish.

Specification of cells in the primitive blood lineage can be traced to their positions during gastrulation in zebrafish (*Kimmel et al., 1990*): ALM macrophages are originally derived from progenitors on the dorsal side of the embryo, and PLM erythrocytes originate from the ventral side of the embryo (*Davidson and Zon, 2004*; *Warga et al., 2009*). Erythrocyte progenitors corresponding to the ALM and PLM are also likely to be positioned on the dorsal or ventral sides respectively in gastrulating *Astyanax* embryos. Therefore, increased erythropoiesis in the ALM suggests that a relative ventralization of hematopoietic development has evolved in cavefish compared to surface fish. This possibility is supported by expanded erythroid development induced by overexpression of the ventralizing factors *bmp2b* and *bmp7* in zebrafish gastrulae (*Lengerke et al., 2008*) and is in line with evidence implicating the importance of maternal factors in dorsal-ventral axis determination of cavefish embryos (*Ren et al., 2018*; *Ma et al., 2018*; *Torres-Paz et al., 2019*). Maternal control of primitive erythropoiesis may promote maximal oxygen distribution at the early stages of development in hypoxic environments. This finding extends our understanding of evolutionary differences in early hematopoiesis to events beginning during oogenesis.

In the present investigation, we have examined primitive hematopoiesis and red blood cells in the Pachón cavefish population. About 30 different *Astyanax* cavefish populations have been described in northeastern Mexico, and some of these cavefish are thought to have been derived independently from surface fish ancestors (*Gross, 2012*; *Herman et al., 2018*). It is conceivable that diverse cavefish lineages inhabit cave pools with different levels of oxygenation. Therefore, in the future it would be interesting to determine the status of erythrocytes in additional cavefish populations. It will also be important to determine if other aquatic cave animals modify their oxygen delivery systems to cope with hypoxia.

Teleosts show considerable diversity in utilizing hemoglobin and red blood cells for oxygen transport during development: some species begin hemoglobin synthesis during embryogenesis, whereas others postpone hemoglobin and erythrocyte production until much later in larval development (*Wells, 2009*). Our studies on the induction of hemolytic anemia with PHZ confirm the functional significance of red blood cells produced during early *Astyanax* development. At the same PHZ concentrations, surface fish, but not cavefish with residual red blood cells, show negative effects on growth, and hypoxia treatment inhibited growth in surface fish but not in cavefish. These results suggest that red blood cells are necessary for oxygen transport at early stages of *A. mexicanus* development and that increased erythrocytes may be advantageous in cavefish. This is contrary to zebrafish, where PHZ induced anemia showed that red blood cells are not required during the first few weeks of development under normoxic conditions (*Pelster and Burggren, 1996*), although under severe hypoxia zebrafish embryos also use red blood cells to increase the supply of oxygen (*Rombough and Drader, 2009*).

Teleosts have evolved different strategies to cope with environments depleted in oxygen, including changes in gill morphology (*Sollid et al., 2003*; *Dhillon et al., 2013*), behavioral avoidance of hypoxic waters (*Congleton, 1980*; *Mandic et al., 2009*), and altered metabolism (*Torres et al., 2012*). *Astyanax* surface fish larvae appear to deal with hypoxia by conserving energy and reducing growth. It is likely that growth is regulated similarly to zebrafish via upregulation of *igfbp1a* (*Kajimura et al., 2005*). The upregulation of *igfbp1* under hypoxia is regulated by HIF1 (*Kajimura et al., 2006*). *Astyanax* cavefish, which are presumably unable to avoid hypoxic waters in their natural environment, show increased expression of some of the *hif1* genes in either hypoxic or normoxic conditions. The constitutive increase in *hif1*-related gene expression may have conferred growth resistance to hypoxia, even under conditions in which *igfbp1* is upregulated, and may be responsible for increased red blood cell number.

The increased expression of the metabolic gene *hk1* in cavefish compared to surface fish indicates an increase in glycolysis, consistent with the observations of *Krishnan et al., 2020*. The increase in glycolysis does not seem to affect the TCA cycle since *ldha* and *pdk1* are not differentially expressed between cavefish and surface fish (*Medley et al., 2020*). Wild captured cavefish also have higher expression of the genes encoding enzymes involved in glutathione metabolism compared to wild caught surface fish (*Krishnan et al., 2020*; *Medley et al., 2020*). Glutathione is an important player in the mitigation of ROS, and the Pentose Phosphate Pathway produces a large portion of the NADPH used in the production of glutathione (*Stincone et al., 2015*). Therefore, the constitutive expression of *hif1*-genes in cavefish could play a role in ROS mitigation via glutathione during hypoxia (*Birnie-Gauvin et al., 2017*).

In conclusion, the present investigation has revealed two potential adaptations that may have permitted *Astyanax* cavefish to cope with hypoxic cave waters. First, cavefish have enhanced erythrocyte development, and as a maternal effect this phenotype may have increased the capacity of cavefish to carry and distribute essential oxygen to tissues and organs early in development. Second, cavefish have adjusted to hypoxic conditions by evolving constitutively high expression levels of some of the *hif1* genes. Together, these changes may be important for the successful adaptation of *Astyanax* to hypoxic subterranean environments.

# Materials and methods

**Key resources table**

| Reagent type (species) or resource | Designation | Source or reference | Identifiers | Additional information |
|---|---|---|---|---|
| Gene (*Astyanax mexicanus* surface fish) | *Hemoglobin subunit beta-2-like, hbb2* | NCBI:GeneID 111196758, Ensembl:ENSAMXG00000031275 | | |
| Gene (*Astyanax mexicanus* surface fish) | *Growth factor independent 1 A transcription repressor a, gfi1aa* | NCBI:GeneID103029320 Ensembl:ENSAMXG00000006669 | | |
| Gene (*Astyanax mexicanus* surface fish) | *LIM domain only 2 (rhombotin-like 1), lmo2* | NCBI:GeneID111190647 Ensembl:ENSAMXG00000032986 | | |
| Gene (*Astyanax mexicanus* surface fish) | *Lymphocyte cytosolic protein, lcp1* | NCBI:GeneID103042114 Ensembl:ENSAMXG00000012855 | | |
| Gene (*Astyanax mexicanus* surface fish) | *hbbe2* | Ensembl:ENSAMXG00005017210 | | |
| Gene (*Astyanax mexicanus* surface fish) | *Ribosomal protein L13a, rpl13a* | NCBI:geneID: 103025160 Ensembl:ENSAMXG00000033532 | | |
| Gene (*Astyanax mexicanus* surface fish) | *Guanine nucleotide binding protein (G protein) beta polypeptide 1b, gnb1b* | NCBI:GeneID 103023031 Ensembl:ENSAMXG00000040710 | | |
| Gene (*Astyanax mexicanus* surface fish) | *Hypoxia-inducible factor one subunit alpha a, hif1aa* | NCBI:GeneID 103022448 Ensembl:ENSAMXG00000039550 | | |
| Gene (*Astyanax mexicanus* surface fish) | *Hypoxia-inducible factor one subunit alpha b, hif1ab* | NCBI:GeneID 103033873 Ensembl:ENSAMXG00000019342 | | |
| Gene (*Astyanax mexicanus* surface fish) | *Hypoxia-inducible factor one subunit alpha like, hif1alike, hif1al* | NCBI:GeneID 103027586 Ensembl:ENSAMXG00000008564 | | |
| Gene (*Astyanax mexicanus* surface fish) | *Hypoxia-inducible factor one subunit alpha like 2, hif1alike2, hif1al2* | NCBI:GeneID 103041845 Ensembl:ENSAMXG00000007272 | | |
| Gene (*Astyanax mexicanus* surface fish) | *Hexokinase 1, hk1* | NCBI:GeneID 103028521 Ensembl:ENSAMXG00000012670 | | |
| Gene (*Astyanax mexicanus* surface fish) | *Glucose-6-phosphate dehydrogenase, g6pd* | NCBI:GeneID 103035433 Ensembl:ENSAMXG00000017509 | | |
| Gene (*Astyanax mexicanus* surface fish) | *Lactate dehydrogenase A4, ldha* | NCBI:GeneID 103047177 Ensembl:ENSAMXG00000032467 | | |
| Gene (*Astyanax mexicanus* surface fish) | *Pyruvate dehydrogenase kinase 1, pdk1* | NCBI:GeneID 103033744 Ensembl:ENSAMXG00000039808 | | |
| Gene (*Astyanax mexicanus* surface fish) | *Insulin-like growth factor binding protein 1 a, igfbp1a* | NCBI:GeneID 103033920 Ensembl:ENSAMXG00000009512 | | |
| Strain, strain background (*Escherichia coli*) | One Shot Top10 chemically competent cells, *E. coli* | Invitrogen | Cat# 404,003 | |
| Biological sample (*Astyanax mexicanus* surface fish) | Surface fish, SF, Texas | Jeffery lab | | |

*Continued on next page*

*Continued*

| Reagent type (species) or resource | Designation | Source or reference | Identifiers | Additional information |
|---|---|---|---|---|
| Biological sample (*Astyanax mexicanus* cave fish) | Cavefish, CF, Pachón, | Jeffery lab | | |
| Antibody | Anti-Digoxigenin-AP Fab fragments (sheep, polyclonal) | Roche | Cat# 11093274910 | 1:5,000 |
| Recombinant DNA reagent | pCRII-TOPO dual promotor vector | Invitrogen | Cat# 45–0640 | |
| Peptide, recombinant protein | Proteinase K | Roche | Cat# 03115887001 | |
| Peptide, recombinant protein | DNaseI I, RNase-free | Thermo-Scientific | Cat# EN0521 | |
| Commercial assay or kit | SYBR Premix Ex Taq (Tli RNaseH Plus) | Takara | Cat# RR420L | |
| Commercial assay or kit | T7 RNA polymerase | Roche | Cat# 10881767001 | |
| Commercial assay or kit | SP6 RNA polymerase | Roche | Cat# 10810274001 | |
| Commercial assay or kit | ReadyMix Taq PCR Reaction Mix | Sigma | Cat# P4600 | |
| Commercial assay or kit | Dig RNA labelling Mix | Roche | Cat# 11277073910 | |
| Commercial assay or kit | Blocking reagent | Roche | Cat# 11096176001 | |
| Commercial assay or kit | BM Purple AP Substrate, precipitating | Roche | Cat# 11442074001 | |
| Commercial assay or kit | SuperScript III First strand synthesis supermix | Invitrogen | Cat# 18080–400 | |
| Commercial assay or kit | SuperScript IV VILO mastermix with ezDNase | Invitrogen | Cat# 11766050 | |
| Chemical compound, drug | Paraformaldehyde | Electron Microscopy Sciences | Cat# 15,710 | |
| Chemical compound, drug | o-Dianisidine | Sigma | Cat# D9143 | |
| Chemical compound, drug | Phenylhydrazine hydrochloride | Sigma | Cat# 114,715 | |
| Chemical compound, drug | Trizol | Life Technologies | Cat# 15596018 | |
| Software, algorithm | Image-J | https://imagej.nih.gov/ij/ | RRID:SCR_003070 | |
| Software, algorithm | JMP Pro 14 | SAS Institute Inc | | |
| Other | CytoOne 24 well plate | USA Scientific | Cat# CC76727424 | |
| Other | Cellstar 12 well cell culture plate | Greiner | Cat# 665,180 | |
| Other | Netwell inserts | Corning | Cat# 3,478 | |
| Other | Hypoxia chamber, ProOx Model P110 | BioSpherix | | |

## Biological materials

*Astyanax mexicanus* surface fish and cavefish were obtained from laboratory stocks descended from collections in Balmorhea Springs State Park, Texas and Cueva de El Pachón, Tamaulipas, Mexico respectively. Fish were raised in a constant flow culture system as described previously (*Jeffery et al., 2000*; *Ma et al., 2021*). Embryos were obtained by natural spawning and reared at 23 °C. Fish

handling and husbandry protocols were approved by the University of Maryland Animal Care and Use Committee (IACUC #R-NOV-18–59) (Project 1241065–1) and conformed to National Institutes of Health guidelines.

## Generation of reciprocal crosses

The offspring of reciprocal crosses were generated by placing a female cavefish and a male surface fish or a female surface fish and a male cavefish together in a tank and inducing spawning by increased feeding and water temperature (*Ma et al., 2021*). Controls were generated by crossing a surface fish male and surface fish female and a cavefish male and cavefish female.

## Quantification of circulating blood cells

The number of circulating blood cells was analyzed in surface fish and cavefish larvae immobilized by treatment with 2 µg ml$^{-1}$ tricaine buffered with Tris to pH 7.0 (Western Chemical, Inc, Ferndale, WA, USA) at 36 hr post-fertilization (hpf) and placed in water on a concave microscope slide. Blood flow was imaged as cells flowed through the Duct of Cuvier (*Isogai et al., 2001*) for 10 s using a stereoscope (Olympus SZX12) or a compound microscope (Zeiss, Axioskop2) with ×50 magnification and a 5 MP, Color, AmScope Microscope Eyepiece Camera (MD500). Recordings were captured with Photobooth on a Macbook computer via USB (OS 10.13.6). Videos were rendered into separate frames in Adobe Photoshop (CC2017, Adobe Inc, San Jose, CA, USA) and processed and analyzed in Image-J (*Schindelin et al., 2012*). After conversion to 32 bit, a data image showing the movement of particles was created by subtracting two frames 0.15 s apart. This data image was used in the plugin TrackMate (*Tinevez et al., 2017*) to analyze the number of moving blood cells in a region of interest (ROI) covering 50% or more of the blood stream over the yolk (150 × 150 pixels). Using the LoG detector, moving blood cells were automatically detected by setting the "blob diameter" to five and the threshold to 2. An automatic quality threshold was used to obtain the number of circulating blood cells and checked against the recording for accuracy. The blood flow of larvae from at least eight different clutches of eggs per morph were analyzed.

## Staining and quantification of erythrocytes

Red blood cells were stained at 36 hpf with 0.6 mg ml$^{-1}$ o-dianisidine (Sigma-Aldrich, St. Louis, MO, USA), 0.01 M sodium acetate (pH 4.5), 0.65% $H_2O_2$ for 15 min in the dark (*Iuchi and Yamamoto, 1983*). The stained embryos were rinsed in phosphate buffered saline (PBS), fixed in 4% paraformaldehyde (PFA), and imaged as described below for in situ hybridization × 50 magnification. The number of blood cells were counted manually in an ROI (150 × 150 pixels) by hand. Larvae from three separate clutches of eggs were used per morph.

## In situ hybridization

In situ hybridizations were performed using probes to the *hemoglobin subunit beta-2-like* (*hbb2*), *growth factor independent 1 A transcription repressor* (*gfi1aa*), *LIM domain only 2* (*lmo2*), or *lymphocyte cytosolic protein 1* (*lcp1*) genes. Genes were cloned from a 24 hpf surface fish cDNA library and 10 hpf, 24 hpf and 30 hpf cavefish cDNA libraries using the pCRII TOPO dual promoter vector (ThermoFisher Scientific, Waltham, MA, USA) transformed into One shot Top10 cells and the following primers: *hbb2* (24 hpf SF; ENSAMXG00000031275: 5'-gcaggacaagtagaaacctcaaagtc-3' and 5'-tttcgtaagggcagagcctaca-3'), *gfi1aa* (24 hpf CF; ENSAMXG00000006669:5'-gaaggtctgcgctcgtgatatt-3' and 5'- agttatccgcggtgt-gaacag-3'), *lmo2* (10 hpf CF; ENSAMXG00000032986: 5'-ggcctctacaatcgagaggaaa-3' and 5' taccaagttg-ccgtttagtttgg-3'), and *lcp1* (36 hpf CF; ENSAMXG00000012855: 5'-aggccttcagcaaagttgatgtg-3' and 5'-ttcaggtcctctgcaccgatatt-3'). DIG-labeled probes were made using SP6 or T7 transcription kits (Roche, Mannheim, Germany) from linearized plasmid or, in the case of *gfi1aa*, from a PCR product made with the cloned cDNA as template and the above-mentioned primers, after a RNA-polymerase promoter site was added to the 5' end.

In situ hybridizations were performed as previously described (*Ma et al., 2014*). Briefly, stored (–20 °C) 4% PFA fixed and methanol dehydrated embryos were rehydrated stepwise into PBS, fixed with 4% PFA, digested with proteinase K, fixed with 4% PFA, and hybridized with probes at 60 °C for 16 hr. Un-hybridized probe was removed by 2 X and 0.2 X SSCT (150 mM sodium chloride; 15 mM

sodium citrate; 0.1% Tween 2) stringency washes followed by incubation in MABT blocking solution (Roche) and subsequently with anti-DIG-AP Fab fragments (Roche). Embryos were rinsed in MABT buffer and PBS, equilibrated in AP buffer, and stained with BM-Purple (Roche). The stained embryos were imaged and photographed using a Zeiss Discovery V20 stereoscope with a Zeiss AxioCam HRc camera.

## Induction of hemolytic anemia with phenylhydrazine

Phenylhydrazine hydrochloride (PHZ; Sigma-Aldrich, St. Louis, MO, USA) was used to induce hemolytic anemia (*Houston et al., 1988*). At 14 hpf, 20 manually de-chorionated embryos were placed in clean fish system-water containing a particular concentration of PHZ or clean fish system water as a control. Embryos were incubated with PHZ for 28 hr when blood circulation was clearly visible over the yolk. Embryos were fixed and used for in situ hybridization with a *hbb2* probe. In a separate experiment, embryos were imaged on a Zeiss Axioskop2 microscope with a Axiocam 503 camera and measurements of tail length (the axial length from the posterior end of the yolk mass to the tip of the tail) were made in Image-J. After imaging, PHZ treated embryos and controls were stained with o-dianisidine and imaged for blood cell quantification as described above. Because low numbers of blood cells were present after PHZ treatment, blood cells were counted in a larger area covering the whole anterior side of the yolk in lateral view (400 pixels wide). Larvae of two separate clutches of eggs were used per morph for the latter experiment.

Surface fish larvae were also treated with PHZ at 34 hpf for 1 hr, rinsed extensively for 1 hr, imaged with a microscope as described above, and left for 16 hr before imaging again. At the end of the experiment, larvae were stained with o-dianisidine. The length of the post-anal tail was measured using Image-J and growth determined as the differential between the two time points. Blood cell number was determined by manual counting as described above.

## Laboratory hypoxia treatment and growth determinations

A hypoxia chamber (ProOx Model P110, BioSpherix, Parish, NY, USA) was used to create a hypoxic laboratory environment in which oxygen was reduced by nitrogen gas (HP, Airgas, Hyattsville, MD, USA). At 30 hpf larvae were exposed to hypoxia (1 mg/L oxygen) for 18 hr or to normoxia (outside of the chamber). Larvae were placed individually in wells of a 24-well plate (CytoOne, USA scientific, Ocala, FL, USA) containing 2 ml clean fish system-water. Twenty-four larvae were used per treatment for surface fish and 48 larvae for cavefish from two separate clutches of eggs per morph. To measure post-anal tail growth, each larva was imaged before placement in a well and again at the end of the treatment using a microscope as described above. Image-J was used to measure post-anal tail length (*Figure 6A*) and growth determined as the differential between the two time points. To measure eye size, eye diameter was measured after the 18 hr. hypoxia or normoxia treatment with ImageJ. After imaging, larvae were stained with o-dianisidine and blood cells were counted from images as described above.

## RNA isolation and quantitative real-time polymerase chain reactions

To compare the extent of early erythropoiesis between cavefish and surface fish, the expression of *hemoglobin beta embryonic 2* (*hbb2*) and *hbbe2* (*hbbe2* encodes an *A. mexicanus* embryonic β-hemoglobin most similar to zebrafish *hbbe1*; *Ganis et al., 2012*) genes and the *gfi1aa* gene were quantified at 10, 24 and 60 hpf. Three potential reference genes were evaluated: *ribosomal protein L13a* (*rpl13a*), *actin alpha 1* (*acta1b*), and *lsm12a*. Only *rpl13a* did not vary over time or between surface and cavefish and was therefore used as the final reference gene. RNA was extracted from 30 embryos with Trizol (ThermoFisher), treated with RNase-free DNase I (ThermoFisher), cleaned and concentrated by phenol/chloroform extraction and precipitated with ammonium acetate and ethanol. Poly(A)-primed cDNA was made with SuperScript III First Strand Synthesis SuperMix (ThermoFisher) and used in qPCR with Takara SYBR Premix Ex Taq (Tli RNaseH Plus) (Takara Bio USA Inc, Mountain View, CA, USA) and LC480 (Roche).

In the laboratory hypoxia experiments, RNA was isolated from 24 larvae placed in the 15 mm Netwell insert (Corning, Corning, New York, USA) in a well of a 12-well plate. Two hours after the start of the hypoxia treatment the larvae were collected in Trizol (ThermoFisher) within 1 min after taking

**Table 2.** Primer sequences used for gene expression analysis with qPCR.

| | | | | |
|---|---|---|---|---|
| rpl13a | GeneID: 103025160 | ENSAMXG00000033532 | caagtactgctgggccacaaag | aggaaagccaggtacttcaatttgtt |
| gfi1aa | GeneID: 103029320 | ENSAMXG00000006669 | agtgtgtgtgatcgaccttcaga | ggacattcttcattgtctggtgacg |
| hbbe2 | | ENSAMXG00005017210* | taaatccctctgcagggctctgat | cctgatcacctccggattagccataata |
| hbb2 | GeneID: 111196758 | ENSAMXG00000031275 | gctcacggtgtagttgttctc | ggatccacgtgcagtttctc |
| gnb1b | GeneID: 103023031 | ENSAMXG00000040710 | ctctgctaaactgtgggatgtg | ccgttagggaagaaacagatgg |
| hif1aa | GeneID: 103022448 | ENSAMXG00000039550 | cagcaccaacacacacactcaa | gtcactgaccaccagtcctaca |
| hif1ab | GeneID: 103033873 | ENSAMXG00000019342 | gcatgggccttacacagttt | gcaccagcatttccctcatt |
| hif1alike | GeneID: 103027586 | ENSAMXG00000008564 | tgcctcacctgcttctaactct | agctgtattctcctctggcttga |
| hif1alike2 | 103041845 | ENSAMXG00000007272 | cattctaagttccagcccatcc | cattggctgcaccatctctc |
| hk1 | GeneID: 103028521 | ENSAMXG00000012670 | ctcaatcggctgaaggacaacaa | agccgtcgagaatactgtggat |
| g6pd | GeneID: 103035433 | ENSAMXG00000017509 | tcctactctgtggtggttgtt | gagacggtctgcttcagtatct |
| ldha | GeneID: 103047177 | ENSAMXG00000032467 | tgtggtgtccaacccagttgata | agcgagctgagtccaagttagt |
| pdk1 | GeneID: 103033744 | ENSAMXG00000039808 | tcctcaaccagcacactcttct | agtgacacgacagtgaggatcaa |
| igfbp1a | GeneID: 103033920 | ENSAMXG00000009512 | cccaacagaagctggaagataag | ctgcccatccagagttgattc |

*blasts to the cavefish genome but also cloned from surface fish in this study.

them out the hypoxia chamber by removing the Netwell inserts from the wells, decanting excess water, and the larvae were immersed in Trizol. RNA was isolated for six samples of 24 larvae each with three samples per clutch of eggs. Two separate clutches of eggs were used per morph.

The amounts of *hypoxia-inducible factor 1* (*hif1aa*) and its duplicate gene *hif1ab*, *hypoxia-inducible factor 1-like* (*hif1alike*) and its duplicate gene *hif1alike2* (equivalent to *hif3a*), *insulin growth factor binding protein 1 a* (*igfbp1a*), *hexokinase 1* (*hk1*), *glucose-6-phosphate dehydrogenase* (*g6pd*), *lactate dehydrogenase a* (*ldha*), and *pyruvate dehydrogenase kinase 1* (*pdk1*) were quantified by qPCR in surface fish and cavefish larvae exposed to hypoxic or normoxic conditions. We evaluated *rpl13a*, *acta1b*, *endoplasmic reticulum protein 44* (*erp44*) and *guanine nucleotide binding protein ß polypeptide 1b* (*gnb1b*) as potential reference genes: *gnb1b* had Ct values closest to the Ct values of the queried genes, did not differ between normoxic and hypoxic surface fish or cavefish, and was chosen as a reference gene. RNA was extracted from 24 embryos and treated with ezDNase and cDNA was made with SuperScript IV VILO mastermix (ThermoFisher). The primers used in qPCR analysis are shown in *Table 2*.

A ΔCt for each gene was calculated by subtracting the average Ct value of each reference gene. For comparison of gene expression in cavefish to surface fish over time, ΔΔCt was calculated by subtracting the average ΔCt of surface fish at 10 hpf from each ΔCt for each gene. For comparison of gene expression under normoxic and hypoxic conditions, the ΔCt of surface fish under normoxic conditions was subtracted from the ΔCt for each gene. For graphical representation, the fold change was calculated as $2^{-(\Delta\Delta Ct)}$, where values > 1 show an increase and values < 1 a decrease. Variation was expressed as the range of fold change $2^{-(\Delta\Delta Ct + stdev\Delta\Delta Ct)}$ for the upper value or $2^{-(\Delta\Delta Ct - stdev\Delta\Delta Ct)}$ for the lowest value.

## Statistics

Statistics were done using JMP pro 14 (SAS Institute Inc). Normal distribution of data was determined using the Shapiro Wilk test. Comparison of blood cell number was done using Wilcoxon/Kruskal-Wallis Rank Sums test (1 F N = 72, 1 G N = 36; p < 0.05). Expression as ΔΔCt of *hbb2, hbbe2,* and *gfi1aa* over time was analyzed per gene with a two-way ANOVA followed by pairwise comparison with Student's t-test (*hbb2* N = 8, *hbbe2* N = 4, *gfi1aa* N = 3; p < 0.05). Comparisons of blood cell number in reciprocal hybridization experiments were analyzed using a Wilcoxon/Kruskal-Wallis Rank Sums test followed by Wilcoxon for each pair (N = 20; p < 0.05). Differences in tail length between surface fish and cavefish at different concentrations of PHZ were analyzed using Wilcoxon/Kruskal-Wallis Rank Sums test followed by Wilcoxon for surface fish (SF) or cavefish (CF) for each pair (SF 0, 2 mg L$^{-1}$ N = 23, SF 1, 3 mg L$^{-1}$ N = 24, CF 0, 1, 2 mg L$^{-1}$ N = 24, CF 3 mg L$^{-1}$; p < 0.05). Comparison of blood cell number at different concentrations of PHZ were analyzed with Wilcoxon/Kruskal-Wallis Rank Sums test followed by Wilcoxon for each pair (SF 0, 1, 2 and CF 3 mg L$^{-1}$ N = 20, SF 3 mg L$^{-1}$ N = 10, CF 0 mg L$^{-1}$ N = 19, CF 1 mg L$^{-1}$ N = 21, CF 2 mg L$^{-1}$ N = 22; P < 0.05). Post-anal tail growth at 0 and 5 mg L$^{-1}$ PHZ was analyzed with a Student's t-test (0 mg L$^{-1}$ N = 19, 5 mg L$^{-1}$ N = 20; P < 0.05) and cell number with Wilcoxon/Kruskal-Wallis Rank Sums test (N = 10; P < 0.05). Post-anal tail growth under normoxic and hypoxic conditions was analyzed with Wilcoxon/Kruskal-Wallis Rank Sums test followed by Wilcoxon for each SF and CF pair (SF-N and SF-H N = 23, CF-N N = 46 and CF-H N = 47; P < 0.05). Cell number under normoxic and hypoxic conditions was analyzed with one-way ANOVA followed by Student's t-test (SF-N N = 22, SF-H N = 21, CF-N N = 47, CF-H N = 45; P < 0.05). Eye diameters were analyzed by one-way ANOVA followed by Student's t-test (SF-N N = 18, SF-H N = 14, CF-N N = 30, CF-H N = 28; P < 0.05). Expression as ΔΔCt of genes under normoxic and hypoxic conditions was analyzed per gene by one-way ANOVA followed by Student's t-test or when variances where unequal with Wilcoxon/Kruskal-Wallis Rank Sums test followed by Wilcoxon for each pair (N = 6; P < 0.05).

## Acknowledgements

This research was supported by NIH grant EY024941 to WRJ. We thank Ruby Dessiatoun and Karina Lacroix for maintenance of the *A. mexicanus* colony and Mandy Ng for technical assistance.

## Additional information

### Funding

| Funder | Grant reference number | Author |
| --- | --- | --- |
| National Institutes of Health | EY024941 | William R Jeffery |

The funders had no role in study design, data collection and interpretation, or the decision to submit the work for publication.

### Author contributions

Corine M van der Weele, Conceptualization, Data curation, Formal analysis, Investigation, Methodology, Resources, Software, Validation, Visualization, Writing - original draft, Writing - review and editing; William R Jeffery, Conceptualization, Funding acquisition, Project administration, Supervision, Writing - original draft, Writing - review and editing

### Author ORCIDs

William R Jeffery http://orcid.org/0000-0002-6997-2946

### Ethics

This study was performed in strict accordance with the recommendations in the Guide for the Care and Use of Laboratory Animals. All of the animals were maintained and handled according to Institutional Animal Care guidelines of the University of Maryland, College Park (IACUC #R-NOV-18-59) (Project 1241065-1). All surgery was performed under anesthesia and every effort was made to minimize suffering.

Decision letter and Author response
Decision letter https://doi.org/10.7554/eLife.69109.sa1
Author response https://doi.org/10.7554/eLife.69109.sa2

## Additional files

### Supplementary files
• Transparent reporting form

### Data availability
All data generated or analyzed during this study are included in the manuscript and supporting file. Source data files have been provided for Figures 1, 2, 4, 5, 6 and 7.

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
