## [Editor Report]

We believe it will have a significant impact on our understanding of cavefish adaptation in particular and adaptation to low oxygen environments in general.

---

## [Decision Letter]

**Decision letter after peer review:**

Thank you for submitting your article "Cavefish adapt to hypoxia by constitutive overexpression of hypoxia-inducible factor genes and increased erythrocyte development" for consideration by *eLife*. Your article has been reviewed by 3 peer reviewers, one of whom is a member of our Board of Reviewing Editors, and the evaluation has been overseen by Didier Stainier as the Senior Editor. The following individuals involved in review of your submission have agreed to reveal their identity: Misty Riddle (Reviewer #2); Rachel Brewster (Reviewer #4).

You will see that our referees find your work intriguing, but have raised a number of important concerns, and in the light of these issues, we cannot offer to publish your paper in *eLife*, at least as it stands. If, however, you feel that you can address the points raised by our referees and make a compelling response to all of the issues raised, then we would be willing to consider a revised version of your paper for possible publication. Given the concerns raised by the referees, we would be looking for additional experimental data along the lines requested to strengthen the evidence in support of the key claims.

Essential revisions:

1) More data is needed to show that CF do better than SF in a low oxygen environment, especially the growth experiments need larger numbers as the results are highly variable.

2) The use of a hypoxia chamber or a comparable set up is needed to alleviate concerns about the reproducibility of the hypoxic values between trials.

*Reviewer #2 (Recommendations for the authors):*

Figure 1: Why wasn't o-dianisidine staining used for the quantification? Figure 1B is not discussed in the main text. I think adding this quantification would strengthen the paper.

Figure 2B: I am not really convinced that HBB2 is expanded in the PBI by the image. There does appear to be staining in the anterior part of the fish however. The expansion in the anterior portion should be mentioned rather than expansion in the PBI.

Line 227. Line 358-361: The authors claim that "cavefish are less sensitive to hemolytic anemia". I don't agree with this interpretation of the data. I would interpret the results as cavefish being less sensitive to developing hemolytic anemia with increasing concentrations of PHZ because they have more blood cells. To compare the effect of reducing blood cells, the authors could compare SF treated with 1 mg/L PHZ to CF treated with 4 mg/L; based on Figure 3A, these treatment groups have about equivalent blood cells. You may conclude from the data that cavefish are more sensitive to a reduction in red blood cells; cavefish treated with 2 mg/L PHZ look equivalent to untreated SF in terms of hbb2 staining, but they have notochord defects and reduction in length. The authors should re-evaluate this experiment and add text to the results considering these points.

Figure 4B: The data should be reported as total number as in figure 1F instead of as a % of untreated. This would help the reader interpret the results of the experiment. The authors should then compare the phenotypes of SF and CF when they have equivalent number of red blood cells. For example, in figure 1D is tail length of CF treated with 4 mg/L significantly different from SF treated with 1 or 2 mg/L? These treatments look like they result in a similar level of red blood cells.

Figure 5B. "the results showed that axial growth was significantly reduced in cavefish in both normoxic and hypoxic conditions relative to surface fish under normoxic conditions." The result shows there is only a difference in the normoxic condition. The authors do not report the statistics comparing SF-normoxic to CF-hypoxic.

Line 361: "Hypoxia treatment inhibited axial growth in SF…". The results are not significant. Authors should increase the number of biological replicates and/or length of exposure.

Line 500: The authors say that 40-50% normoxic was lethal. Did they compare lethality in SF and CF embryos? I think it would be of interest to include lethality as a measurement in the study. Are there concentrations that CF can survive while SF die? This may be a more relevant measurement than growth considering whether the increase in red blood cells is adaptive.

More data is needed to show that CF do better than SF in a low oxygen environment and that if you decrease HIF expression and/or erythrocyte number this advantage is eliminated. Experimentally increasing HIF expression and/or erythrocyte number in SF and showing that this gives them an advantage in low oxygen water would provide additional strong evidence that the trait is adaptive.

The authors should add text to the discussion that the study only focused on one of the cavefish populations and that examining the other populations would be an interesting future direction. The authors should add "Astyanax mexicanus" to the title. There are many different species of cavefish and this may not be a general feature for how cavefish adapt to hypoxia.

*Reviewer #4 (Recommendations for the authors):*

1) The use of other assays (e.g metabolic rate under hypoxia) to test the functional relevance of increased erythrocyte production and enhanced transcription of hif1 would strengthen the conclusions.

2) In other teleosts, it is well established that elongation of the embryo is driven by convergent extension movements in the axial mesoderm. It would therefore be worthwhile examining whether this process occurs differently in cavefish relative to surface fish under hypoxia.

3) Hif protein levels under hypoxic conditions should be measured.

4) The use of a hypoxia chamber to control oxygen levels is recommended.

---

## [Author Response]

Reviewer #2 (Recommendations for the authors):Figure 1: Why wasn't o-dianisidine staining used for the quantification? Figure 1B is not discussed in the main text. I think adding this quantification would strengthen the paper.

Thank you for this comment. We agree and have added quantification of o-dianisidine stained cells in revised Figure 1G. These results strengthen our conclusion that cavefish have more red blood cells than surface fish.

Figure 2B: I am not really convinced that HBB2 is expanded in the PBI by the image. There does appear to be staining in the anterior part of the fish however. The expansion in the anterior portion should be mentioned rather than expansion in the PBI.

We also thank the reviewer for this comment. In the revised manuscript, we have now emphasized the expansion of *hbb2* expression in the ALM and PLM rather than the PBI. The comment also prompted us to rewrite these results for clarification. There are two major conclusions of our experiments: One is that hematopoietic/erythropoietic domains are present in both the anterior and posterior regions of *Astyanax* embryos, which is not the case in zebrafish and other teleost embryos, where they are present only posteriorly, and another is that both the anterior and posterior hematopoietic/erythropoietic domains (ALM and PLM) are expanded in cavefish compared to surface fish based on expression of the *hhb2*, *gfi1aa,* and *lmo2* genes. The section of the revised manuscript was rewritten to emphasize on these two conclusions. Please see Lines 140-166. We also revised Figure 2 (now Figure 3) to highlight the two major conclusions of this section of the manuscript

Line 227. Line 358-361: The authors claim that "cavefish are less sensitive to hemolytic anemia". I don't agree with this interpretation of the data. I would interpret the results as cavefish being less sensitive to developing hemolytic anemia with increasing concentrations of PHZ because they have more blood cells. To compare the effect of reducing blood cells, the authors could compare SF treated with 1 mg/L PHZ to CF treated with 4 mg/L; based on Figure 3A, these treatment groups have about equivalent blood cells. You may conclude from the data that cavefish are more sensitive to a reduction in red blood cells; cavefish treated with 2 mg/L PHZ look equivalent to untreated SF in terms of hbb2 staining, but they have notochord defects and reduction in length. The authors should re-evaluate this experiment and add text to the results considering these points.

We agree with the reviewer, but point out that the major purpose of this section of the manuscript was to determine whether erythrocytes have a role in early *Astyanax* development. We thought that this section was necessary because older literature from the zebrafish community suggested that erythrocytes are produced during early embryogenesis but are not required for oxygen transport until much later in larval development. Thus, it was necessary investigate whether red blood cells are required for early development in *Astyanax*. We believe that our PHZ results, which are now fortified by additional experiments, answer this question in the affirmative. In particular, the experiments in revised Figure 5D show that larval growth is suppressed in the absence of red blood cells. We feel that these results strongly support the conclusion that erythrocytes are functional and required during early *Astyanax* development.

Because of other concerns listed in the Reviewer’s comment immediately above, and also taking into account the recommendation to report red blood cell phenotypes by number suggested in the comment immediately below, we have repeated all of the experiments in question. The results are shown in revised Figure 5C, which compares larval tail length and red blood cell number at increasing PHZ concentration over a 28 hr time span. The findings are as follows. First, red blood cell numbers are decreased in concert with increasing PHZ concentration in both surface fish and cavefish. Second, the loss of red blood cells is more abrupt and severe in surface fish than in cavefish. Third, increasing PHZ concentrations reduce tail length in surface fish but not in cavefish, which have more erythrocytes. We interpret these results to mean that cavefish are less sensitive to hemolytic anemia because they have more blood cells, matching the conclusion of this reviewer.

We could not compare PHZ-treated surface fish and cavefish with the same number of red blood cells because we could not find a PHZ concentration with closely matching erythrocyte numbers in SF and CF that would allow this comparison to be made (Please also see the comments immediately below and Figure 5C).

Lastly, we have re-written the section of the Results describing these data. Please see Lines 183-207.

Figure 4B: The data should be reported as total number as in figure 1F instead of as a % of untreated. This would help the reader interpret the results of the experiment. The authors should then compare the phenotypes of SF and CF when they have equivalent number of red blood cells. For example, in figure 1D is tail length of CF treated with 4 mg/L significantly different from SF treated with 1 or 2 mg/L? These treatments look like they result in a similar level of red blood cells.

We agree with the reviewer. In revised Figure 5C the data is shown as the total number of red blood cells. However, as pointed out above, we could not compare PHZ- treated surface fish and cavefish at equivalent red blood cell numbers because there was no PHZ concentration where approximately equivalent numbers could be found.

Figure 5B. "the results showed that axial growth was significantly reduced in cavefish in both normoxic and hypoxic conditions relative to surface fish under normoxic conditions." The result shows there is only a difference in the normoxic condition. The authors do not report the statistics comparing SF-normoxic to CF-hypoxic.

We have now repeated these results using a hypoxia chamber to better control oxygen levels, and have confirmed significantly different effects of hypoxia tolerance between SF and CF. Please see revised Figure 6. These results support the conclusion that CF are less sensitive to hypoxia than SF.

Line 361: "Hypoxia treatment inhibited axial growth in SF…". The results are not significant. Authors should increase the number of biological replicates and/or length of exposure.

Please see our comment immediately above. In revised Figure 6, we show significant differences in axial growth between cavefish and surface fish exposed to hypooxygenation in a hypoxia chamber.

Line 500: The authors say that 40-50% normoxic was lethal. Did they compare lethality in SF and CF embryos? I think it would be of interest to include lethality as a measurement in the study. Are there concentrations that CF can survive while SF die? This may be a more relevant measurement than growth considering whether the increase in red blood cells is adaptive.

Thank you for this comment. We agree and during the interval between submissions we have performed many experiments to compare survival of surface and cavefish under hypoxic (in a hypoxia chamber) and normoxic conditions. Unfortunately, none of these experiments produced interpretable data. During the course of these experiments it became clear that successful survival studies will require much more extensive preliminary work to develop appropriate conditions using many different oxygenation levels, which was not possible during the interval allowed for revising this manuscript.

More data is needed to show that CF do better than SF in a low oxygen environment and that if you decrease HIF expression and/or erythrocyte number this advantage is eliminated. Experimentally increasing HIF expression and/or erythrocyte number in SF and showing that this gives them an advantage in low oxygen water would provide additional strong evidence that the trait is adaptive.

Thank you for this advice. We have now performed new experiments to address these concerns using a hypoxia chamber. The results show that growth (and erythrocyte number) in cavefish is significantly less sensitive to oxygen reduction than in surface fish (revised Figure 6). Also, to further address these concerns, we have now performed an experiment in which erythrocytes were completely eliminated and this significantly reduced axial growth (revised Figure 5D).

The authors should add text to the discussion that the study only focused on one of the cavefish populations and that examining the other populations would be an interesting future direction. The authors should add "Astyanax mexicanus" to the title. There are many different species of cavefish and this may not be a general feature for how cavefish adapt to hypoxia.

We agree with the reviewer that examining other cavefish populations would be interesting and have now added the suggested material to the Discussion. Please see Lines 314-321.

The expansion of the title information in the Summary, which would appear just below the title in a published manuscript, clearly states that the work was done in the cavefish *Astyanax mexicanus*. Therefore, in the interest of keeping an already long title in check, we have not added the genus and species name to the manuscript title.

Reviewer #4 (Recommendations for the authors):1) The use of other assays (e.g metabolic rate under hypoxia) to test the functional relevance of increased erythrocyte production and enhanced transcription of hif1 would strengthen the conclusions.

Thank you for this suggestion. We agree that confirmatory information might be obtained by different assays under hypoxia conditions. However, we point out that the two assays we have conducted in the revised manuscript, tail and eye growth, using the hypoxia chamber permit the conclusion that cavefish have greater tolerance to hypoxia than surface fish, and that this phenotype is likely to be a consequence of more erythrocyte development. In the time allotted for revision, we have repeated all of the hypoxia experiments with these two criteria as assays. We agree that, with additional time, adding more assays might provide further confirmatory evidence, but would not strengthen our present conclusions beyond the present evidence.

2) In other teleosts, it is well established that elongation of the embryo is driven by convergent extension movements in the axial mesoderm. It would therefore be worthwhile examining whether this process occurs differently in cavefish relative to surface fish under hypoxia.

We agree that it would be interesting to see if hypoxia affects notochord morphogenesis. However, convergent extension movements forming the notochord occur during and shortly after gastrulation in zebrafish and *Astyanax*. The experiments on hypoxia in our manuscript began at the 13-14 somite stage, considerably later than convergent extension in the developing chordamesoderm. Thus, we believe that the suggested experiments, although interesting, would have little direct significance for our current study.

3) Hif protein levels under hypoxic conditions should be measured.

We agree that this would be an excellent topic for future research. However, at present we do not have the tools (antibodies, etc) in the *Astyanax* system to specifically detect, distinguish and quantify the four Hif proteins. Our manuscript is focused on *hif1* transcripts, which can be detected using established qPCR procedures, and our conclusions do not go beyond the transcriptional level. We also point out that HIF system activation has been previously documented in a variety of fish species based entirely on (indirect) studies of *hif* expression levels. The article by Mandic et al. (2021) in our reference list reviews these studies. We believe that this approach is also appropriate for our study, particularly since our only claim is that *hif* transcription is overexpressed by hypoxia.

4) The use of a hypoxia chamber to control oxygen levels is recommended.

Thank you again for this excellent suggestion. During the interim between submissions, we purchased a hypoxia chamber and using it repeated all of our hypoxia experiments, as we have described above. The results are shown in revised Figures 6 and 7. Our basic conclusions are still the same, but this approach has permitted us to make stronger conclusions about the differential effects of hypoxia on surface fish and cavefish in the revised manuscript.